# Image Sequence Analysis via GRU and Attention for Trachomatous Trichiasis Classification

**Juan C. Prieto**[1]                  JPRIETO@MED.UNC.EDU
[1] *Department of Psychiatry, UNC Chapel Hill, NC*
**Hina Shah**[1]                    HINASHAH@EMAIL.UNC.EDU
**Kasey Jones**[2]                   KRJONES@RTI.ORG
[2] *Center for Data Science, RTI International, Research Triangle Park, NC*
**Robert F. Chew**[2]                RCHEW@RTI.ORG
**Hashiya M. Kana**[3]              HASHIYA@EMAIL.UNC.EDU
[3] *Department of Epidemiology, UNC Chapel Hill, NC*
**Jerusha Weaver**[3]              JWEAVE25@EMAIL.UNC.EDU
**Rebecca M. Flueckiger**[4]         RFLUECKIGER@RTI.ORG
[4] *Global Health Division, RTI International, Research Triangle Park, NC*
**Scott McPherson**[4]            SMCPHERSON@RTI.ORG
**Emily W. Gower**[3,4]             EGOWER@UNC.EDU

**Editors:** Under Review for MIDL 2021

## Abstract

Chlamydia trachomatous is an infectious ocular condition that can cause the eyelid to turn inward so that one or more eyelashes touch the eyeball, a condition call trachomatous trichiasis (TT), which can lead to blindness. Community-based screeners are used in rural areas to identify patients with TT, who can then be referred for proper medical care. Having automatic methods to detect TT will reduce the amount of time required to train screeners and improve accuracy of detection. This paper proposes a method to automatically identify regions of an eye and identify TT, using photographs taken with smartphones in the field. The attention-based gated deep learning networks in combination with a region-identification network can identify TT with an accuracy of 91%, sensitivity of 92% and specificity of 87%, showing that these methods have the potential to be deployed in the field.

**Keywords:** Trachoma, Trachomatus Trichiasis, image classification, attention based networks

## 1. Introduction

Trachoma is the leading infectious cause of blindness world-wide(Resnikoff et al., 2004; Flueckiger et al., 2019). It is transmitted from person to person through ocular and nasal secretions or eye-seeking flies carrying the infection from one person to another. Repeated infections result in eyelid scarring, causing it to turn inwards and the eyelashes to touch the eye, a condition called trachomatous trichiasis (TT). If not corrected, inturned eyelashes can abrade the cornea, which can lead to blindness (West et al., 2006). Surgery is the primary method for treating TT.

The majority of individuals with TT live in low- and middle-income countries where access to health services is limited. Surgical campaigns are conducted to provide surgery through outreach services. To facilitate planning and align resources, individuals with TT are often identified in advance and invited to the upcoming surgical camp. Current methods for TT case identification are suboptimal. Often, local community members are asked to serve as case screeners in their communities. These case screeners receive a brief training on how to identify TT and then are asked to go door-to-door in their village to screen for TT. These case screeners often have limited success at identifying TT appropriately, with positive predictive values for case identification ranging from 15-30%(Greene et al., 2015). The time and cost associated with door-to-door case finding is substantial(West, 2013), and engaging case screeners with ophthalmology expertise to conduct this work is cost and resource-prohibitive in many scenarios.

In this paper, we propose a method to close the gap between experts and non-experts in assessing TT by increasing screening accuracy, and reducing the time required for training community-based screeners. Previous work has demonstrated that early signs of trachoma that present on the underside of the eyelid can be detected better than chance(Kim et al., 2019). However, to our knowledge, we are the first to classify trichiatic eyelashes, a later stage of the disease, using machine learning. We identify TT cases through attention-based recurrent neural networks using automatically segmented regions of the eye. We chose this method based on experiments performed on our data set as well as additional constraints of computational cost for the algorithm, (*i.e.*, it must be possible to run on a mobile device in remote communities without reliable internet connectivity).

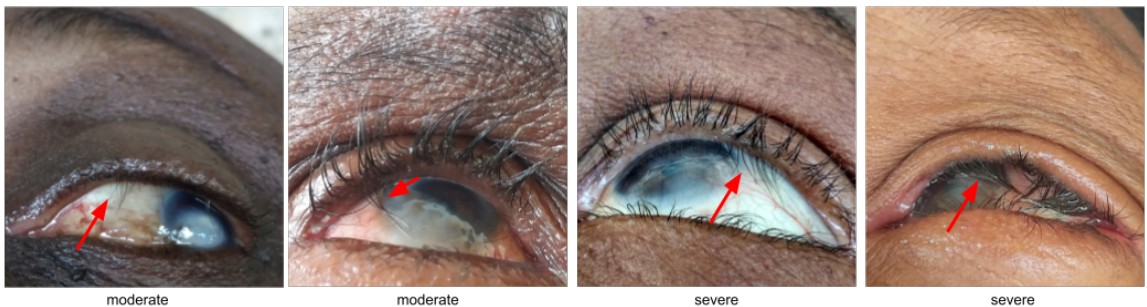

Figure 1: Examples of eyes with moderate TT (left two) and severe TT.

## 2. Data

We use images that were collected as part of the Maximizing Trichiasis Surgery Success Trial (MTSS)(Bayissasse et al., 2020) in Ethiopia. The MTSS trial was designed to evaluate three surgical approaches for correcting TT. A total of 5001 individuals with previously unoperated TT were enrolled and followed for one year. Individuals were required to have TT in at least one eye. At each study visit, images were taken of each upper eyelid, with the participant looking up, using either a Motorola Moto X Pure edition or a Samsung Galaxy 8 smartphone. After imaging, a trained study examiner evaluated each eyelid for the presence of TT, defined as one or more eyelashes touching the eye or evidence of eyelash

removal through epilation. The examiner recorded the number and location of any trichiatic eyelashes in the study database. Images were transmitted to the coordinating center at University of North Carolina at Chapel Hill, where they were assessed by a certified photograph grader. For the work in this paper the grader marked images with moderate or severe TT (based on number of eyelashes touching the eye, but no epilation) as having TT, and the eyes without TT as normal/healthy. Figure 1 shows examples of moderate and severe TT. In addition to images of eyes with TT, we collected upper eyelid images of 1,121 adults without TT, from the same region in Ethiopia. The combined dataset has 6,048 images. From these, our grader marked the sclera, cornea and upper eyelid regions of the eye for 1,113 images using a custom extension to 3D Slicer[1] (Kikinis et al., 2014)[2]. Figure 2 shows examples of the expertly segmented regions of the eye. A set of 1,706 images with good quality were also identified for the task of classification.

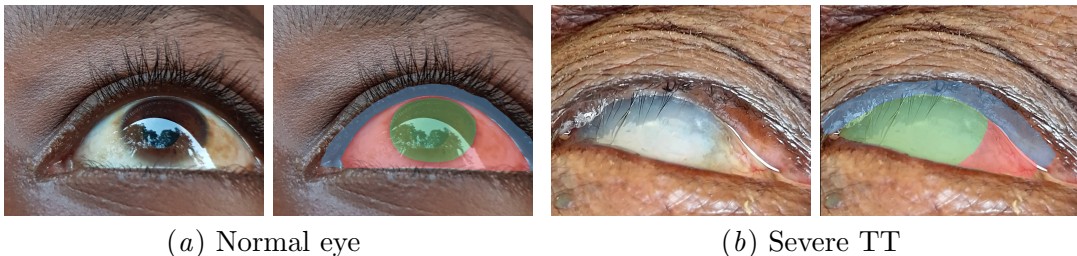

$(a)$ Normal eye $\qquad\qquad$ $(b)$ Severe TT

Figure 2: Training samples and corresponding label maps generated by an expert rater. The sclera, cornea, and upper eyelid are shown in red, green and blue respectively.

## 3. Related work

Machine learning is becoming an increasingly popular means to screen for eye diseases and disorders. Although, certain eye conditions, such as diabetic retinopathy (Asiri et al., 2019) and glaucoma (Barros et al., 2020), have received considerable attention in the literature, far less progress has been made on Neglected Tropical Diseases (NTDs), such as trachoma despite TT impacting an estimated 2.8 million people worldwide (Flueckiger et al., 2019). (Kim et al., 2019) performed a multi-step process to distinguish images of trachomatous inflammation from normal eye images. The models were trained on images taken of participant's everted upper eyelid, labeled as follicular inflammation (TF), intense inflammation (TI), or normal using the WHO simplified trachoma assessment system(Thylefors et al., 1987). They used a multi-layer perception classifier to identify eyelid pixels followed by a shallow convolutional network to detect trachoma. They found the less severe TF designation to be more difficult to identify consistently than TI.

Examples of detecting regions of the eye are seen in (Rot et al., 2018) and (Luo et al., 2019). While (Luo et al., 2019) identify the pupil and sclera regions of the eye in widely varying images using a shape constrained network, (Rot et al., 2018)'s method labels six different regions of the eye including the eyelashes using an encoder-decoder scheme. These

---

1. www.slicer.org

2. https://slicer.readthedocs.io/en/latest/user_guide/modules/segmenteditor.html

methods are a) trained on images of healthy eyes, and b) do not identify the eyelid region, which our method needs for TT identification.

In general, a neural network (NN) can be trained either from scratch, or by adding a few additional layers (Convolution, RNN, dense, etc.) to pretrained models to customize for the classification or regression task. However, it is known that using pre-trained models instead of training models from scratch leads to equal or superior performance (Zhou et al., 2017). In our experiments, we use pre-trained weights (on the ImageNet(Deng et al., 2009) data set) from VGG19(Simonyan and Zisserman, 2014), ResNet50 (He et al., 2016), MobileNetV2 (Sandler et al., 2018) NN architectures. The pre-trained weights from VGG19 have been used to develop state-of-the-art classification (Su et al., 2015; Kanezaki et al., 2018; Ma et al., 2018; Cho et al., 2014).

## 4. Methods

In this section we describe our analysis pipeline to detect if an eye has TT or not. Our initial experiments to do binary classification using labeled TT and non-TT images directly were unable to perform well, despite fine tuning several models. Hence, we focused on a procedure by using areas around the upper eyelid (where human experts also diagnose TT according to WHO protocols). As such, our final pipeline includes a segmentation, extraction, and a concluding classification step.

We first train a residual UNET(Ronneberger et al., 2015; Zhang et al., 2018) to identify cornea, sclera and upper eyelid regions of the eye in both healthy and TT-affected eyes. The segmented upper eyelid region is then used to extract a sequence of upper eyelid images at full resolution to train a TT classifier (ttAttNet). We first describe our pre-processing and then explain ttAttNet.

Our pre-processing pipeline (ttUNETCrops) is shown in Figure 3(a). The main components are the segmentation of the eye's regions of interest (ROI) via UNET, and forming an image sequence by cropping image regions around the upper eyelid. We use the segmentation network to locate our region of interest (i.e. upper eyelid) and focus the analysis on this region. By doing so, we can perform the analysis at maximum resolution. The sequence of image crops is created by fitting a curve in the upper eyelid ROI and sampling it uniformly between $[x_{min}, x_{max}]$, and extracting 32 contiguous cropped images around the eyelid ROI.

Figure 3(b) shows the sequence analysis pipeline (ttAttNet). ttAttNet uses a bi-directional recurrent neural network (RNN) with Gated Recurrent Units (GRU)(Cho et al., 2014), and Bahdanau additive attention layers(Bahdanau et al., 2014, 2016). The attention layers allow focusing on specific frames of the input sequence by computing weights that highlight the importance of certain frames and reduce the contribution of others for the final network decision. Moreover, these weights may be used for visually explaining predictions to stakeholders or for additional analysis at a later stage.

We start by processing each frame of the sequence with a feature extraction and a global average pooling layer in order to extract salient features. Thus, we reduce each frame in the sequence (384x384 pixels) to a vector of (512 for VGG19, 2048 for ResNet50, 1280 for MobileNetV2) features. Recurrent neural networks (RNN) with Long-short term memory (LSTM)(Hochreiter and Schmidhuber, 1997) or GRUs units can then analyze this sequence

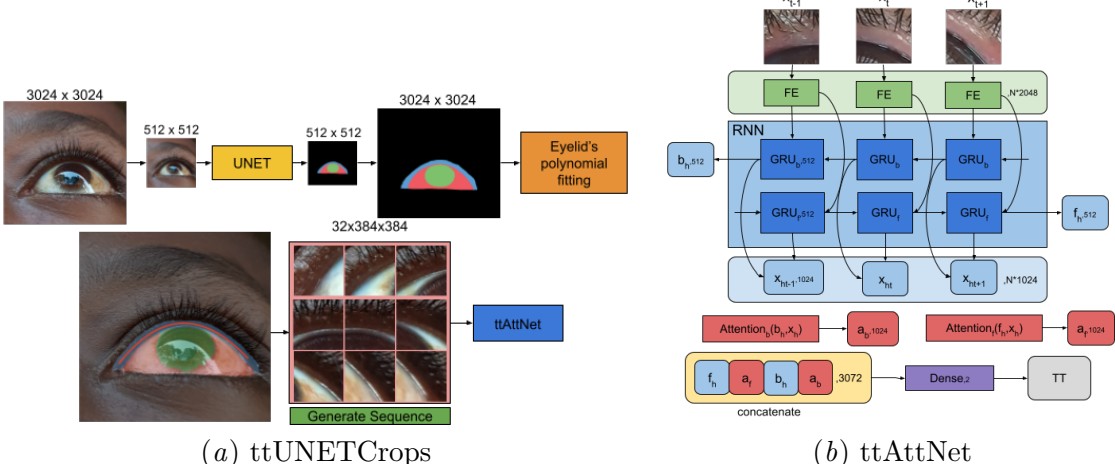

$(a)$ ttUNETCrops $\qquad\qquad\qquad (b)$ ttAttNet

Figure 3: a) Image sequence creation at full image resolution. The segmentation produces three labels for the sclera (red), cornea (green), and upper eyelid (blue). The upper eyelid ROI is used to fit a smooth curve or polynomial (red line) which is used to guide the extraction of crops and form an image sequence. The sequence is analyzed by (ttAttNet). b) Feature extraction (FE) (VGG19, ResNet50 and MobileNetV2) for each frame in the input sequence. Analysis using Bi-directional Recurrent Neural Network (RNN) with Gated Recurrent Units (GRU) and attention layers. The attention layers allows focusing on relevant frames of the sequence.

of 1-dimensional features. GRUs have been reported to have a slightly better performance than LSTM. LSTM networks also have more parameters and take longer to train (Fu et al., 2016; Khandelwal et al., 2016).

We hypothesize that using images at lower resolution is not ideal for TT classification since TT classification often requires detecting a few eye lashes pointing in the wrong direction. In other words, these fine image details may be lost if the images are down sampled, which is a common pre-processing step to use state of the art neural network architectures for image classification. Nevertheless, customizing existing NN architectures to handle images at larger resolution is possible, but it may not be computationally efficient. On the contrary, in this paper, we propose to use existing architectures for image analysis and focus only on the upper eyelid region using image patches at the highest resolution level.

We compare the performance of our approach (ttUNETCrops+ttAttNet) with various other approaches. For comparison we use: 1. Re-sampled images at 512x512, and VGG19, ResNet50, and MobileNetV2 as feature extraction (output size of feature extraction is 16x16x512, 16x16x2048, 16x16x1280 respectively). 1.1 ttVGG19$_{512}$: Features from VGG19 and training an additional 2DConv layer (filter size 3x3, stride 2x2, 512 units), and 3 fully connected (FC) layers (the 3 FC layers are part of the original VGG19 architecture). 1.2 ttResNet50$_{512}$ - ttMobileNetV2$_{512}$: Features from ResNet50/MobileNetV2

and an additional 2DConv (filter size 3x3, stride 2x2, 2048/1280 units) and a FC layer. 2. Our approach which re-samples an input image to 512x512, segments the image to locate the upper eyelid and extract a sequence of 32 images of size 384x384 at full resolution (ttUNETCrops), followed by ttAttNet which analyzes the sequence of features extracted with VGG19, ResNet50, and MobileNetV2 (output size of feature extraction is 1x512, 1x2048, 1x1280 as we use a global average pooling layer on each). The ttAttNet architecture is the same for all experiments as is shown in Figure 3(b).

## 5. Results

We select a set of 1,706 images (996 TT, 709 non-TT) images with good quality around the eyelid region for classification. Our grader marked the sclera, cornea and upper eyelid regions of the eye for 1,113 images. These are somewhat different sets of images, with an overlap of 455 images that have both manual segmentations and TT grading. From this overlap set we randomly chose 308 images to test our approach. In summary, 73% for training and 27% for testing the UNET segmentation, and 82% for training and 18% for testing the classification. A validation set is randomly chosen (10%) from the training set during training of each network. We use the validation set to stop the training using the early-stopping criteria and avoid over/under-fitting.

### 5.1. ttUNETCrops

The residual UNET was trained using the images segmented by our expert grader as ground truth. The training went for 124 epochs, the learning rate was set to $1e-4$, dropout rate was set to 0.15, Adam optimizer and a categorical cross entropy loss function to discriminate between 4 classes (background, sclera, cornea, and upper eyelid).

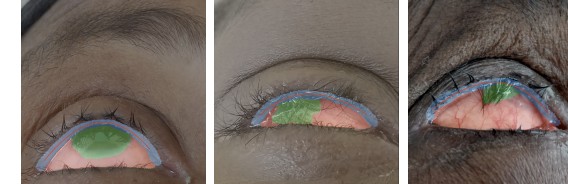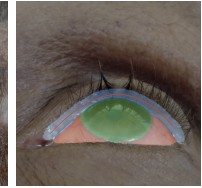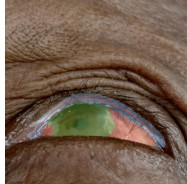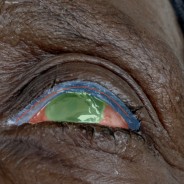

Figure 4: Segmentation results for randomly chose samples in our data set. The fitted curve guides the extraction of contiguous image crops.

Table 1 shows the results for the image segmentation task. The dice score for the eyelid, which is the most important ROI for our experiment, is 0.71. This is an acceptable outcome for the next phase of the analysis and it will be demonstrated by our classification results. Figure 4 shows the segmentation plus polynomial fitting for randomly chosen samples in our data set.

### 5.2. ttAttNet

In our experiments, we trained different neural networks to evaluate the performance of the classification task. Table 5.2 shows the result of the classification task for all the networks

| Label | Precision | Recall | F1-score | Dice |
|-------|-----------|--------|----------|------|
| Background | 0.99 | 0.98 | 0.99 | 0.98 |
| Sclera | 0.85 | 0.80 | 0.83 | 0.82 |
| Cornea | 0.85 | 0.91 | 0.98 | 0.88 |
| Upper Eyelid | 0.70 | 0.73 | 0.71 | 0.71 |

Table 1: Accuracy of segmentation task

| NN | input | class | precision | recall | f1-score | accuracy |
|----|-------|-------|-----------|--------|----------|----------|
| $ttVGG19_{512}$ | 16x16x512 | normal | 0.84 | 0.72 | 0.78 | 0.83 |
| | | tt | 0.81 | 0.91 | 0.86 | |
| $ttResNet50_{512}$ | 16x16x2048 | normal | 0.90 | 0.73 | 0.81 | 0.85 |
| | | tt | 0.82 | 0.94 | 0.87 | |
| $ttMobileNetV2_{512}$ | 16x16x1280 | normal | 0.87 | 0.70 | 0.78 | 0.82 |
| | | tt | 0.79 | 0.92 | 0.85 | |
| $ttAttNet_{VGG19}$ | 32x512 | normal | 0.78 | 0.84 | 0.81 | 0.87 |
| | | tt | 0.92 | 0.88 | 0.90 | |
| $ttAttNet_{ResNet50}$ | 32x2048 | normal | 0.87 | 0.86 | 0.87 | 0.91 |
| | | tt | 0.92 | 0.93 | 0.93 | |
| $ttAttNet_{MobileNetV2}$ | 32x1280 | normal | 0.83 | 0.81 | 0.82 | 0.87 |
| | | tt | 0.89 | 0.91 | 0.90 | |

Table 2: Classification results for different architectures and feature extraction methods. $ttVGG19_{512}$, $ttResNet50_{512}$ and $ttMobileNetV2_{512}$ use the features extracted with the respective networks as inputs of images re-sampled to 512x512. No pooling operation is applied. The ttAttNet networks use features extracted for each of the 32 frames (384x384) in the sequence and a global average pooling.

evaluated. $ttAttNet_{VGG19}$, $ttAttNet_{ResNet50}$ and $ttAttNet_{MobileNetV2}$ trained for 4, 7, 5 epochs respectively. Learning rate was set to $1e-4$, Adam optimizer and a categorical cross entropy loss function to discriminate between normal and tt is used.

Our approach of classifying a sequence of cropped images around the ROI of upper eyelid outperforms the classification of full images, even if different feature extraction schemes are used, with ResNet giving an overall accuracy of 91%. The weights computed by the attention layers may be used for further analysis or to visualize the most relevant frames. Figure 5 shows the top 3 frames selected by the attention layer.

## 6. Conclusions

In this work, we have proposed an approach to analyze high resolution images of eyes captured by commercial phones, and identify TT - an infectious disesase of the eye. By focusing the analysis only to the region of interest, we have demonstrated that we can

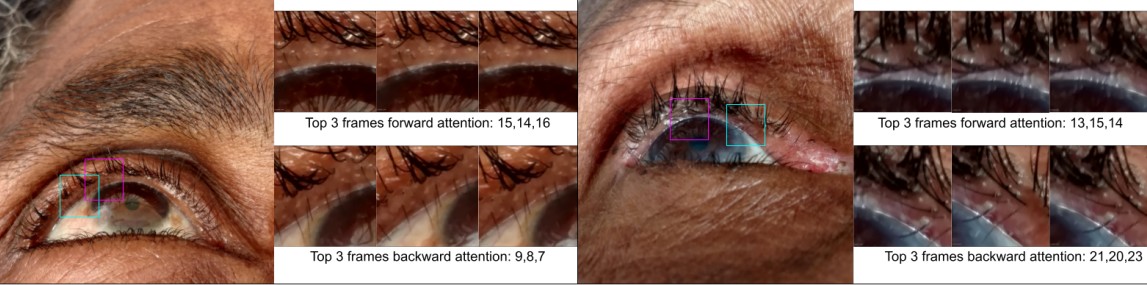

Figure 5: The weights produced by the attention layers may be used for visualization and validation. The figure shows the top 3 frames selected by the forward (magenta) and backward (cyan) attention layers. Both images show eyelashes pointing inwards towards the eye detecing TT. Both examples were misclassified by the ttVGG19$_{512}$ NN.

achieve an f1-score of 0.93 using full resolution images vs. 0.87 when the images are re sampled to 512x512. This pipeline is to be deployed on mobile devices; hence, there are computational constraints and the trained models need to be as lean as possible without compromising the accuracy of the classification. Re-sampling the images to 512x512, using state-of-art neural networks for feature extraction and training an additional few layers yields a higher number of parameters and lower accuracy than the proposed approach. The experiments in this paper demonstrate that using image samples at full resolution is needed to have the best possible accuracy for TT prediction, as it may come down to identifying a single eyelash pointing in the wrong direction. Section 7.2 shows a summary of the number of parameters for each neural network used in this paper. The attention weights can be used to display which frames of the sequence contribute to the final decision of the pipeline. The results from the attention modules used in ttAttNet indicate that anomalies in the eyelid are a great indicator to classify normal v.s. TT images. Down sampling the images to enable other architectures results in lower accuracy. During our analysis, we also found that image quality, resolution, and TT severity played a role in the accuracy of the classification. A standardized protocol for taking the photos is followed, which includes using similar quality cameras at same distances from the patient. In future, we will assess the robustness of our models when the protocol is not followed. Our future work will also create classification models to predict severity of TT, include mild cases of TT, as well as images with epilation.

This paper reports the first step in demonstrating that machine learning is an achievable approach to TT identification. The MTSS trial has demonstrated that smartphone-naïve individuals can be trained how to navigate a simple app and take high-quality images of the human eyelid. Future steps will require employing the methods described here into a smartphone-ready app and then assessing app functionality in the field. Ultimately, this line of work has the potential to impact the global approach to TT case finding by increasing screening accuracy and subsequently getting patients who need TT management access to sight-saving services.

## Acknowledgments

We are grateful to the many study participants in Ethiopia who allowed us to collect images of their eyelids. Additionally, we would like to thank RTI International and the National Eye Institute for the funding that made this work possible.

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

## 7. Appendix

### 7.1. Early stopping and number of epochs

The number of epochs for our experiments are set based on the early stopping criteria, *i.e.*, after the validation loss stops improving. The number of epochs for ttVGG19$_{512}$, ttResNet50$_{512}$ and ttMobileNetV2$_{256}$ is set to 4, 27, 6.

### 7.2. Number of parameters for each neural network

| NN | Parameters | Total |
|---|---|---|
| $\text{VGG19}_{Fextraction}$ | 20,024,384 | - |
| $\text{ResNet50}_{Fextraction}$ | 23,587,712 | - |
| $\text{MobileNetV2}_{Fextraction}$ | 2,257,984 | - |
| $\text{ttVGG19}_{512}$ | 153,362,432 | 173,386,816 |
| $\text{ttResNet50}_{512}$ | 37,752,832 | 61,340,544 |
| $\text{ttMobileNetV2}_{512}$ | 14,748,160 | 17,006,144 |
| ttUNETCrop | 14,160,352 | - |
| $\text{ttAttNet}_{VGG19}$ | 6,303,748 | 40,488,484 |
| $\text{ttAttNet}_{ResNet50}$ | 11,022,340 | 49,358,212 |
| $\text{ttAttNet}_{MobileNetV2}$ | 8,663,044 | 25,081,380 |

Table 3: Number of parameters for each architecture. The total includes the parameters from the feature extraction + the additional layers trained for each architecture. For the ttAttNet rows, it includes the parameters from ttUNETCrop as well.

