# OpenReview forum: "Image Sequence Generation and Analysis via GRU and Attention for Trachomatous Trichiasis Classification"
_MIDL.io/2021/Conference — MIDL 2021_

### Official Review · AnonReviewer4 · 2021-02-28

**Confidence:** 3
**Preliminary Rating:** 3
**Recommendation:** Poster

**Summary:**

The authors propose a method to classify good quality images of upper eyelid as either normal
or having (TT) Trachomatous Trichiasis.
The steps in the proposed method are 1. A ttstack net (UNet) for segmenting the  cornea, sclera and upper eyelid regions of the eye in both healthy and TT-affected eyes. 2. Polynomial curve fitting using the upper lid labels as the Region of Interest used to generate a sequence of 32 images of various portions of the eyelid, each 384x384 sized. 3. A set of VGG19 layers with global average pooling that takes these image sequences and gives feature vectors of length 512 each. 4 tt-AttNet - a set of bi-directional recurrent neural networks (RNNs) with Gated Recurrent Units (GRU) and attention units that analyze these feature vectors and provide weights for visualization of regions where the eyelid had turned inwards.
5. The forward and backward hidden layer output vectors each of length 512, the forward and backward attention layer outputs each 1024, all put together 3072-vector is fed to fully connected layers - two layers with 4096 units and last layer with two units to classify as TT or not.

**Strengths:**

1. This interesting work targets diagnosis of an important problem namely, trachoma, one of the  Neglected Tropical Diseases, which is an important problem since it leads to adverse conditions like blindness and also it is contagious.

2. The proposed methodology considers that the network needs to be lightweight to be able to deploy on mobiles.

3. The proposed method uses simple modules like GRU yet effective in terms of computations as compared to the vanilla CNNs or pure multi-layer perceptrons or earlier work on the same problem.

4. The introduction and motivation of the problem is well-written.


**Weaknesses:**

The cornea will be affected by TT and eventually causes blindness. Hence the information about cornea also needs to be taken into account to classify the severity of the disease. In addition to this, this disease could also affect the lower eyelid. Under such cases the ROI images and the direction of the turned eyelids would be different. The proposed method must generalize well for both the eyelids. Neither the information about the cornea or the lower (instead of upper, in case) eyelid is taken into consideration for obtaining attention.

“In this work, we focus on images of upper eyelid with good quality, and that are graded as either normal or having moderate or severe TT without epilation.” -- this gives an impression that the proposed method classifies images as normal eye, moderate TT or severe TT.  However the output dense layer has only 2 units classifying as normal or TT.

It is mentioned in the paper that polynomial fitting on the eyelid region is done to guide the generation of image sequences. It is not clear how the fitted curve is used to extract eyelid portion for further analysis.

For the methods used for comparison, directly the overall images are used as inputs but for the classification part of the proposed method the ROI images (upper eyelid) go as inputs. Hence for the baseline methods also ROI images should be considered.

Since only the upper eyelid is the region of interest, a justification of why there is a need to segment all the three structures vs segmenting only the eyelid must be specified in the experiments section.  In other words, would segmenting all the three regions help in precise segmentation of the upper eyelid must be justified.





**Deanonymize Review:**

no

**Detailed Comments:**


1. Figure 3b could be made readable in the following aspects:
The yellow block must be labelled as a concatenation operation. There are two unlabelled arrows in input of the forward and backward paths of the bidirectional GRU blocks.

2. One suggestion is that in page 5, in the illustrations about Figure 3b, which is one of the key works of this paper, the order of sentences could be refactored. The sentences are all relevant but lack continuity.
For instance if the paragraph talks about how data flows from VGG to the GRUs and then to the attention, there would be more clarity.

3. Using fully connected layers as the terminology instead of dense layers would add more clarity. This is because the word “dense” is commonly used when referring to densely connected neural networks.

**Justification Of The Preliminary Rating:**

The problem is interesting and there are scopes towards extending the work for detecting early stages of TT.

LSTM’s and GRU’s can be found in speech recognition, speech synthesis, text generation and to generate captions for videos. The use of these units for this application is appreciated.

**Paper Type:**

validation/application paper

**Questions To Address In The Rebuttal:**

1. The authors use ttVGG19_{256} and ttVGG19_{512} models for performance comparison. Are the sequence images of the upper eyelid given directly to these models or the input snapshot image?

2. There could be conditions wherein the cornea and the eyelashes camouflage, especially under varying illumination conditions and the cornea is in the eyelid region. For instance, in the case of images which are of good resolution yet poor illumination in the reference paper, “Deep Multi-class Eye Segmentation for Ocular Biometrics” by  Rot et al., 2018, what would be the performance of the proposed approach?

3. How does the curve guide the eyelid image sequences? In other words, points on the curve are chosen and a region around the point is taken as part of the eyelid region?


**Special Issue:**

no

---

> ### Author Response · Authors · 2021-03-18
> **Clarification of inputs for the classification networks and requirements for the approach**
>
> We thank the reviewer for taking the time to review our work, and their feedback.
>
> 1. The sequence of images is not used in ttVGG19_{256} and ttVGG19_{512}. The inputs to these networks are the cropped images around the eye resampled to 256x256 and 512x512 and using the features extracted with VGG19. In the final version of the paper, we have removed the 256x256 experiment and report only on the 512x512.
> We test other feature extraction networks such as ResNet50 and MobileNetV2. The objective of these comparisons is to demonstrate that there are more relevant features for TT classification such as eyelashes pointing in the wrong direction. By focusing the analysis on this region, we achieve better accuracy than using the resampled images and using networks such as VGG19, ResNet50, etc.
> We also point out that using an input image of 512x512 using feature extraction and adding a few additional layers for training/fine-tuning puts the number of parameters above the proposed approach. This initial experiment led us to split our analysis into two steps. First, identify the region of interest. Second, focus the analysis on this region and use images at the highest possible resolution. Although ttVGG19, ttResNet50, ttMobileNetV2 use the resampled images as input and may rely on image features from the cornea/sclera for the final decision of the network, we demonstrate that focusing only on the eyelid yields higher accuracy.
>
> 2. Our data set contains images of varying lighting conditions, we expect that our approach will handle such environment conditions without a problem.
>
> 3. This is correct, the polynomial fitted to the eyelid ROI and is uniformly sampled between [Xmin, Xmax], crops size 382x382 are extracted around the sampled points.
>
> To answer some of your concerns:
>
> We did not use the Cornea region in the current state of the work since we followed The WHO recommendation of identifying TT based on the presence of eyelashes.  While it is ultimately the cornea that is damaged and results in blindness, the cornea is not taken into consideration when assessing whether an individual has TT or not. To address the concern of not using the lower eyelid: the vast majority of TT occurs in the upper eyelid.  Only 5-10% of individuals who develop upper eyelid TT also develop lower eyelid TT, and it is extremely rare that TT affects only the lower lid.  The WHO defines TT based strictly on the upper eyelid. So, we have followed their guidance in focusing on the upper eyelid.  Further, in order to assess whether eyelashes are touching (Based on a picture) the camera lens needs to be angled in such a way that the lash-globe contact is visible.  It is not feasible to try to assess both the upper and lower eyelids in the same picture. Ultimately, the method could be used for lower eyelids, but separate images would need to be taken.  We see no problems with translating this approach to the lower eyelid.  However, as noted above, TT primarily affects the upper eyelid, so the lower eyelid is a second priority for a later date.
>
> We have only 2 decisions from the classification because this was the aim of the primary field deployment.
> The image sequence is created by extracting images around sample points using the polynomial that is fit to the eyelash ROI.
>
> For justifying the eyelash segmentation: while cornea and pupil are a standard feature in the image, the eyelid is not, and we see a lot of variation in the sizes, shape, and even existence of it. Having a segmentation for the region of the eye aids in finding the eyelid, and also saying if there is an eye in the image or not. (For some individuals, the cornea and eyeballs are non-existent).
> We have also updated our evaluation section for classification in the paper, as we compare various architectures on resampled full images, and image sequences around the upper eyelid.

---

### Official Review · AnonReviewer1 · 2021-03-05

**Confidence:** 4
**Preliminary Rating:** 2
**Final Rating:** 3

**Summary:**

This work focuses on the classification of eye images among four different categories, No/Moderate/Sever Trachomatous Trichiasis (TT). The authors mention some difficulties when training a standard CNN for classification with their data, and switch to an alternative approach in which they first segment the eye area, then split it into a series of sequential crops (this would be ttStackNet), and finally a second model to classify the entire image based on this series of crops, using a combination of features extracted from a VGG and a GRU module (this would be ttAttNet).

**Strengths:**

The problem addressed in this paper is unusual and of a high societal impact, given the prevalence of this disease in low-resource areas of the world. The problem identified by the authors (training on downsampled images) leads to an interesting question of how to address training when only a small area of the image contains the relevant information, and we cannot use downsampling due to how fine that information is (the eyelids).

**Weaknesses:**

I find several design decisions in this system to be a bit questionable, even more considering the aim of the authors to deploy this on smartphones. In addition, the evaluation seems to me a bit weak also. Please find below a description of my concerns:

1) It is mentioned in the second section that the available dataset has been labeled for no/moderate/severe TT, and also segmented into sclera/cornea/upper eyelid/background. This sounds like a very rich set of annotations, but when one goes and sees the experiments, they really do not math the expectations. For example, from Table 2, it seems that the authors are training their model to perform binary classification between normal/TT eyes. Wouldn't it be more useful to do 4-class classification? Also, for the segmentation, the authors train a UNet for 4-class segmentation, which is nice. But then if we see the classification model, it never uses the information of which pixels belong to each area. The obtained segmentation is used to locate the eyelid region and manually extract a series of sliding crops following it. These sequences of crops are the only input to the classification stage, discarding the other information present in the segmentation. What is the reason for this procedure? Wouldn't it be more reasonable to simply segment the eye from the background, and then pass the segmented eye (possible stacked with the segmentation) to a CNN so that it can classify it? If the authors are concerned about the image resolution, they could always use a sliding window approach.

2) The disconnection between segmentation and classification is also somehow concerning, in my opinion. First, since one can expect that good segmentations would lead to better classification and bad segmentation could have catastrophic results, the authors should have preserved the same training/test set partitions for segmentation and classification. Unfortunately this is not the case, as they mention that the segmentation model is trained on a 80/20 partition of the data, and the classification is trained/validated following a 10-fold cross-validation scheme.

3) Next, it seems to me that ttStackNet is actually a UNet followed by extraction of a series of crops following the eyelid. I think this does not deserve to be labeled as "ttStackNet", which gives the sensation to the reader that this is a new architecture or something similar. I believe this should be referred to simply as image pre-processing: something like "we extract an eye segmentation before going ahead with the classification, and this segmentation is learned with a UNet. Following, a sequence of crops around the eyelid is extracted and used as input for classification."

4) The baselines used for comparison are a bit unfair in my opinion. They consist of a VGG that takes as input the original image, resizes it to 512x512 or 256x256, and fine-tune the last few layers. A more fair baseline would be to use the segmentation obtained in the previous step to separate the eye from the background, and then train a CNN for classifying the resulting segmented eye. Even better would be to train a joint model that performs segmentation + classification.

5) The selected architectures are really a weird design decision. A standard UNet contains over 30 million parameters, and VGGs are known to be very heavy models that can be easily outperformed by lightweight architectures, even the smaller ResNets. Considering that this model is to be deployed in smartphones, the authors should justify much better their decision to use such heavy architectures by showing benefits when compared to lighter alternatives. They alread mention MobileNet in their discussion, but I believe this architecture should be included in the current version of the paper.

6) Regarding the evaluation of the classifier, why not following the same (better) procedure as for segmentation and use a held-out test set to assess performance? If there is a reason for this, at least the authors should report the mean or median results of the 10-fold CV procedure, with standard deviations of the errors.


**Deanonymize Review:**

no

**Detailed Comments:**

I believe the above list is already quite detailed? By the way, I find the title a bit misleading. Nowadays, when one reads "image generation", we normally expect some kind of generative modeling. However, in this case it refers to segmenting the eye area and extracting crops around the eyelid.

**Final Rating Justification:**

The authors have corrected their evaluation to use a held-out test set, and now include more suitable architectures in their experiments. I am increasing from Weak Reject to Weak Accept because of this, but I am still not 100% sure about acceptance.

**Justification Of The Preliminary Rating:**

I believe this paper needs more work in terms of experimentation to justify the selection of super-heavy architectures to solve a problem, particularly considering the final aim of deploying the system on smartphones.

**Paper Type:**

validation/application paper

**Questions To Address In The Rebuttal:**

I am sorry but at the very least I would need to see a complete restructure of the experiments and evaluation. First by splitting the data in train/val/test for segmentation and following the same splits to perform classification. And second by testing lighter architectures for both the segmentation and classification problems, in order to justify the choices of UNet and VGG-Net.

**Special Issue:**

no

---

> ### Author Response · Authors · 2021-03-18
> **Restructuring of experiments with common hold out test for segmentation and classification and using lighter architectures for testing the approach**
>
> We thank the reviewer for their time to review the paper, and their feedback.
>
> 1. We trained the segmentation to extract different regions of the eye with the hope that a) it will create visual feedback to the screener, and b) we can use this information in the future to improve the classification method. For the purpose of the current state of work, we focus on the areas around the eyelid region to use eyelash information in compliance with the WHO protocol. We followed The WHO recommendation of identifying TT based on the presence of eyelashes.  While it is ultimately the cornea that is damaged and results in blindness, the cornea is not taken into consideration when assessing whether an individual has TT or not.  Trichiatic eyelashes result from a modification of the eyelid margin (it turns inward). So, it was crucial to look at the eyelid region as a first step. As you will see in the results, we did try to just use the region around the segmented eye, and then do classification, but the accuracies are still low.
>
> 2. Thank you for bringing out this point. We restructured our dataset, so that now we have a consistent test set of 308 images for segmentation and classification, and have added the corresponding segmentation accuracies and classification results in the paper
>
> 3. We wanted to highlight the sequence generation. But you are right, it gives a feeling that this is a new method. We have renamed that section to reflect that we have used a preexisting method.
>
> 4. In our initial experiments we did extract the eye, and just ran classification on those images. However, the classification accuracies were always low. The results for these experiments were not included due to space constraints. The comparisons in the paper to other approaches are to demonstrate that there are features in the image that are more relevant for the accurate classification of tt, i.e., eyelashes pointing in the wrong direction.
> Such fine details are lost when downsampling the images. In our approach, this is exactly what we are trying to accomplish: by segmenting the region of interest and focusing the analysis only on this region, we extract crops at the maximum possible resolution. We point to the table in the appendix and the number of parameters for each network. If we want to use the full resolution images (which are cropped around the eye) and range from 2600x2600 to 3200x3200 pixels we will not be able to fit a single model that does segmentation/classification. Adding a few additional layers to the outputs of ResNet50 which include a Conv2D layer with 2048 units, 2x2 stride, 3x3 filter size, a Global Average Pooling layer and a Dense layer already puts the number of parameters above the proposed approach and this is using images re-sampled at 512x512. We decide to split the analysis into two processes, i.e., Identifying the eyelid region and then focus on this area. Our comparisons do not intent to minimize the importance of other classification approaches but instead, to highlight the importance of splitting the analysis into two separate processes and focus the analysis only on the eyelid region. Although ttResNet50 and others may be relying on image features of the cornea/sclera for the output of the classification, we demonstrate that better classification results are obtained by focusing only on the eyelid region. We have now compared ResNet50, VGG19, and MobileNetV2. Since the submission of the paper, we deployed the segmentation and feature extraction steps on a phone and we expect to have the full pipeline in the near future.
>
> 5. We have addressed this concern as mentioned earlier.
>
> 6. The results reported in the initial version of the paper were the mean across all folds. However, for the current version of the paper, we report the segmentation and classification results only on the holdout test set.

---

### Official Review · AnonReviewer3 · 2021-03-07

**Confidence:** 5
**Preliminary Rating:** 1
**Final Rating:** 1

**Summary:**

The study is motivated by the application of deep neural networks in rural areas to facilitate the automated screening of Trachomatous Trichiasis (TT). While the work has a high potential for real world impact, the manuscript in its current form presents several issues due to the lack of proper use of machine learning terminology as well as model selection and evaluation.

**Strengths:**

A good motivation and high potential to make an impact in real life.

Decent practical solutions to combat problems faced during the study.

Use of a both segmentation and classification networks in a pipeline.

A unique problem that goes beyond the well-known examples from ophthalmology, such DR detection, AMD classification, glaucoma detection, etc...

**Weaknesses:**

The title is misleading. "Image Sequence Generation" does not mean "Fixed-size patch extraction from images". Such misleading phrases appear multiple times throughout the manuscript. However, there is not a single "generating process" or "generative model".

Training and model selection:
ttStackNet:
"The data set was split in 80% for training and 20% testing. The number of epochs was set to 30 based on a preliminary study using early-stopping as criteria."
In fact, this means that the model was trained on 80% of data and the "test" partition was essentially the validation set, on which early-stopping was applied w.r.t. a "criteria" that is not mentioned here. Thus, there was no "test" set in the first place. Criteria could also refer to metric such as accuracy, ROC-AUC, etc...
Please, use a proper model development and evaluation strategy here. Also, consider an external validation (out-of-distribution) dataset to estimate the true generalisation performance of your models. This should give you a better idea for your model(s) to be used out there in the field. Include uncertainty as well. These should help evaluate the reliability of predictions from a network.

I am afraid the same mistake was repeat with the second part of the pipeline: ttAttNet.

Apart from these major concerns, data description is not clear to me. In total there are 1121 adults and 11617 images. 1161 images were selected w.r.t. some sort of quality criteria and "mild" examples were excluded. Then, the expert annotated 1113 images from 8089 images. So, if we assume that 48 images were mild, then what is up with 8089? Also, Why were the images with epilation excluded? This should be clarified.

After all, the goal is to develop a DNN-based diagnostic tool feasible for mobile solutions. VGG is a large network and its computational burden is too much, compared modern alternative. The authors are aware of this and propose to use MoblieNet. In this regard, I am a bit surprised to see an unfinished "product/study/manuscript" submitted for peer review.

**Deanonymize Review:**

no

**Detailed Comments:**

Abstract:
a condition call"ed" trachomatous trichiasis (TT)

Introduction:
"Trachoma is the leading infectious cause of blindness world-wide (Resnikoff et al., 2004)." In 2021, is this still the case?  Any up-to-date reference?

The last three paragraphs need to be revised and possibly merged. The first are overlapping, also.

Related work:
"They used a multi-layer perception classifier to identify eyelid pixels followed by a shallow convolutional network to detect trachoma." Is this accurate? Typically, a convolutional stack extracts features and fully connected layers perform the end task, e.g., classification.

Methods:
"the generation of an image sequence." This phrase combined with GRU makes think very different things.

5.2 ttAttNet
One sentence paragraphs look bad. Maybe, use a list or re-work them.

6. Conclusion
"image analysis of high resolution images" Yes. It is all about images but one of them is redundant here.

**Final Rating Justification:**

The study has definitely a potential for impact. However, I still find the manuscript below average due to its structure, writing/presentation and convoluted use of machine learning terminology, despite the revisions.

Unfortunately, I did not share the excitement of their clinical collaborator while reading the manuscript.

**Justification Of The Preliminary Rating:**

Community-based screeners achieve a positive prediction rate of 15-30%, according to the reference given in the paper. This is low bar. With a decent approach, current machine learning algorithm, especially DNNs can surpass this level. This would not be surprise to me. In this regard, I am convinced by the results from an unfinished product evaluated with improper strategies.

On another note, I have no intention of discouraging the authors from developing this work further. It just need to be better than this. I appreciate their efforts for providing accurate care in rural areas.

**Paper Type:**

validation/application paper

**Questions To Address In The Rebuttal:**

Please consider the items in the Weaknesses section.

**Special Issue:**

no

---

> ### Author Response · Authors · 2021-03-18
> **Clarification for data set and split for training/testing**
>
> We thank the reviewer for their time to review the paper and their feedback.
>
> We have changed the title of the paper. Thank you for pointing out the confusion in the title.
>
> As recommended, we made a consistent test set for segmentation and classification tasks that has 308 images, and repeated the training for segmentation and classification tasks. We had to rework our dataset a little bit, so you will see slightly different numbers in the data description and results. Unfortunately, we do not possess out-of-distributions/outliers in our data set. Our 2 sets for the classification task are balanced and should not bias the training of the algorithms.
>
> To answer your questions for rebuttals:  We redid the classification training with different networks: ResNet50, MobileNet, and VGG19. We have updated our results in the pdf version of the paper.  Since the submission of the paper, we have worked on the app development for deploying this pipeline. The requirement for our initial study is to have a near-real-time identification. Our initial runs show that the time taken to extract features (the main bottleneck step of the pipeline) for an input image takes anywhere between 623ms to 8s with MobileNetV2 being fastest, and VGG19 being slowest. We use TensorFlow’s default quantization for compression.
>
> Data Description: Images with epilation were excluded so that we could first train the model on trichiatic eyelashes.  Our next step will be to add in epilation. The dataset of 11,617 images includes 1,691 upper eyelid images of healthy adults. Out of the remaining 9,926 images, 2,137 images have epilation (hence are excluded from this study), 2443 are normal, 3442 images have mild TT (excluded for this study and will be included in future work), and the remaining1903 images have moderate to severe TT. Our grader has marked for us good quality images for classification that are not extremely blurry in the regions with eyelashes. This created a set of 1,706 images for training and testing the classification. Please note that the set of images for training and testing segmentation is a different set and is not limited to only good-quality images. The segmentation was trained/tested on 1,113 images. Although these are somewhat different sets of images, we do have an overlap between them, and our consistent test data of 308 images have both manual segmentations (included in the 1,113 images) and TT grading (1,706 images), these 308 images are left out for testing. The classification and segmentation tasks were approached as being two different tasks, and hence two different sets of images. We have clarified this in the paper, and have used numbers only for the images that were considered for the study.
>
> We have also updated our references and considered your notes in the detailed comments. The comment related to the related work( Kim et al) is accurate – we refer you to look at their original manuscript by Kim et al. They have two steps and hence two networks: the first one identifies the pixels that belong to the eyelid based on the intensity distribution as part of their preprocessing phase, and the second one classifies this set of pixels as having trachoma or not.
>
> This is a product that is exciting to our clinical collaborator and is now targeted for field testing in the near future.

---

### Official Review · AnonReviewer2 · 2021-03-09

**Confidence:** 5
**Preliminary Rating:** 3
**Recommendation:** Poster

**Summary:**

The authors propose an automated machine learning algorithm for a novel clinical problem, the classification of an infectious ocular condition called chlamydia trachomatous. The suggested pipeline consists of a segmentation network (ttStackNet) and a subsequent classification network (ttAttNet). The former network, ttStackNet, is modeled as a U-Net, whereas ttAttNet is a bi-directional GRU with attention layers. Promising results are reported.

**Strengths:**

The paper is clearly written and easy to understand.

This seems to be the first machine learning work that addresses an important but less known clinical problem: classifying trichiatic eyelashes.

The evaluation dataset contains more than 1500 images.

The model choice is reasonable given the hardware constraints (deployment on smartphones).

Sensible ablation studies to compare the influence of the image resolution and justify the model choice.

**Weaknesses:**

The technical contributions of this paper are incremental since the chosen models are well known from the literature (U-Net, bi-directional GRU, attention). The main strengths are the motivation of the clinical problem and the display of a working solution.

It is not quite clear if the split in training and validation set without modeling a test set may have contributed to model overfitting. I don't regard this as a major problem because the dataset is quite large and even a small drop in performance (if at all) would not diminish the selling point of the paper, which is the practical impact of this work.

**Deanonymize Review:**

no

**Justification Of The Preliminary Rating:**

I think this work is technically solid, addresses a novel clinical problem, and is clearly written. The model choices are reasonable and it seems that this work will result in a practical application.

However, it does not offer a novel technical solution and could include further ablation studies to justify the model choices.

**Paper Type:**

validation/application paper

**Questions To Address In The Rebuttal:**

What is the performance gain using attention layers?

How much do bi-directional GRUs improve on uni-directional GRUs?

Have the authors considered ResNets as an alternative to VGG?

Do the authors consider compressing the model using distillation, Bayesian Deep Learning, or any other SOTA compression method?

**Special Issue:**

no

---

> ### Author Response · Authors · 2021-03-18
> **Additional pre-trained networks for feature extraction and comparison**
>
> We thank the reviewer for their time to review the paper and their feedback.
> Following multiple reviewers’ suggestions, we made a consistent test set for segmentation and classification tasks that has 308 images and redid the training with different networks: ResNet50, MobileNetV2, and VGG19. We have updated our results in the pdf version of the paper.
> To answer your questions for rebuttals:
> As shown in the results section, the attention layer allows us to focus the analysis on specific frames of the sequence and increase their contribution to the final decision of the network. There is a performance gain by using the attention layers, but besides the performance gain, we show 2 examples where the attention layer’s top picks are frames characteristic of TT with lashes pointing in the wrong direction. The frames picked by the attention layer will also provide additional feedback to the screener during the examination. The order of the sequence matters when using GRUs or LSTM networks; by including the bi-directional layer, we minimize the importance of the order. Additionally, by using forward and backward direction in combination with the attention layers, we can focus on different parts of the sequence. As shown in the results, the forward and backward layers are picking frames at different locations in the eyelid. We have now also tried ResNet50 and MobileNet in our pipeline, and while MobileNet gives a similar accuracy as VGG19, ResNet50 seems to be performing better. Since the submission of the paper, we have worked on the app development for deploying this pipeline. The requirement for our initial study is to have a near-real-time identification. Our initial runs show that the total time taken to extract features (the main bottleneck step) takes somewhere between 623ms to 8s depending on the feature extraction method, with MobileNetV2 being the fastest and VGG being the slowest. We use TensorFlow’s default quantization for compression.

---

### Meta-Review · Area_Chair1 · 2021-03-28

**Recommendation:** Accept (Poster)

**Metareview:**

After reading the original manuscript, the reviewers' comment, the authors' rebuttal and the new submission, I will follow the opinion of the majority and recommend to borderline accept this paper. I agreed with the reviewers that a better evaluation strategy and a better/rigorously description of the data and the method needed to be done. The authors addressed this in their rebuttal. However, there are still concerns about their acceptance although not clearly justified. Hence, my recommendation.

**Paper Type:**

validation/application paper

---

### Decision · Program_Chairs · 2021-03-31

Accept